# Improving Question Answering over Knowledge Graphs with a Chunked Learning Network

Zicheng Zuo [1], Zhenfang Zhu [1], Wenqing Wu [2], Wenling Wang [3], Jiangtao Qi [1] and Linghui Zhong [1,*]

1 School of Information Science and Electrical Engineering, Shandong Jiao Tong University, Jinan 250104, China; zuozicheng1997@163.com (Z.Z.)
2 School of Economic and Management, Nanjing University of Science and Technology, Nanjing 210094, China
3 Chinese Lexicography Research Center, Lu Dong University, Yantai 264025, China
* Correspondence: 205068@sdjtu.edu.cn

**Abstract:** The objective of knowledge graph question answering is to assist users in answering questions by utilizing the information stored within the graph. Users are not required to comprehend the underlying data structure. This is a difficult task because, on the one hand, correctly understanding the semantics of a problem is difficult for machines. On the other hand, the growing knowledge graph will inevitably lead to information retrieval errors. Specifically, the question-answering task has three difficulties: word abbreviation, object complement, and entity ambiguity. An object complement means that different entities share the same predicate, and entity ambiguity means that words have different meanings in different contexts. To solve these problems, we propose a novel method named the Chunked Learning Network. It uses different models according to different scenarios to obtain a vector representation of the topic entity and relation in the question. The answer entity representation that yields the closest fact triplet, according to a joint distance metric, is returned as the answer. For sentences with an object complement, we use dependency parsing to construct dependency relationships between words to obtain more accurate vector representations. Experiments demonstrate the effectiveness of our method.

**Keywords:** question answering; knowledge graph embedding; chunked learning network





## 1. Introduction

Large-scale knowledge graphs like Freebase [1], DBPedia [2], Yago [3], and NELL [4] contain many facts from the real world, which makes question answering based on knowledge graphs (referred to as KGQA) a vital task. A complex data structure and a large number make it difficult for ordinary users to obtain a large amount of useful knowledge. KBQA (Knowledge Base Question Answering) and KGQA are both related to question-answering systems, but they focus on different types of knowledge repositories. In KBQA, a question is posed in a structured query (such as SPARQL [5]), and the goal is to retrieve the precise answer from the structured knowledge base. The system needs to understand the question, map it to the appropriate entities and relationships in the knowledge base, and retrieve the relevant information to provide an accurate answer. KGQA, on the other hand, extends beyond traditional knowledge bases and deals with more flexible and dynamic knowledge graphs. Knowledge graphs are also structured representations of information but are more expressive and allow for richer relationships and contextual information. They are often represented using semantic web technologies, such as the Resource Description Framework (RDF) or property graphs. Knowledge graphs can incorporate data from various sources and are more capable of representing complex and interconnected knowledge. In KGQA, the question-answering system needs to understand the question, navigate the knowledge graph, and perform more sophisticated reasoning to arrive at the correct answer. KGQA systems often employ natural language processing techniques, graph-based reasoning, and deep-learning methods to handle the complexities of the knowledge graph.

For example, consider the question, "Which actors have won an Oscar and also starred in a Christopher Nolan movie?" To answer, the KGQA system would need to reason through the knowledge graph, identifying entities related to actors, Oscar awards, and Christopher Nolan movies to find the correct answer.

The accurate understanding of question semantics and the effective filtering of interfering information are essential for successfully answering questions in KGQA. Currently, commonly used methods rely on semantic parsing [6–9] and information retrieval. The core concept behind semantic parsing is the conversion of natural language into a sequence of formal logical forms. Through the bottom-up analysis of logical forms, a logical form that can express the semantics of the entire problem is obtained, and the corresponding query sentence is used in the knowledge graph. This method is based on a relatively simple statistical method and has a greater dependence on data. Most importantly, it cannot map the relationship from natural language phrases to complex knowledge graphs. Secondly, supervised learning is needed when obtaining answers. And we need to train a classifier to score the generated logical form. To train such a powerful semantic parsing classifier, a great deal of training data is necessary. Whether it is Freebase [1] or WebQuestion [6], these two datasets have relatively few question and answer pairs. To address this issue, Zhang et al. [10] proposed a structural information constraint, which applies the structural information of the problem to path reasoning based on reinforcement. Zhen et al. [11] adopted a complementary approach, integrating a broader information retrieval model and a highly precise semantic parsing model, eliminating the need for manual template intervention.

The information retrieval method [12–14] is used to extract entities from the question and then search for the entities in the knowledge graph to obtain entity-centric subgraphs. Any node or edge in the subgraph can be a candidate answer. By observing the question and extracting information according to certain rules or templates, the feature vector of the question is obtained and a classifier is established. Then, the candidate answers are filtered by the feature vector of the input question to obtain the final answer [15]. However, KGQA needs to perform a multi-hop search to obtain the target entity when faced with missing inference chains. This makes the time and space complexity of the algorithm grow exponentially.

In addition, the same word can have different meanings in different contexts. We call this phenomenon entity ambiguity. For example, the meaning of an apple in Cook's hand and an apple in Newton's head are completely different. In daily life, people are used to using abbreviations instead of full names, such as Newton instead of Isaac Newton. This causes the algorithm to obtain a narrower entity search space. The diversity of predicates will produce a broader entity search space. When the same predicate connects different entities, its representation will be different, which requires the algorithm to be more robust.

We solve the above difficulties in two ways: (1) By embedding entities and relationships into the same vector space as the knowledge graph, we can naturally solve the problems caused by abbreviations, because similar entities can learn the same vector representation. And entities in different contexts will also obtain different vector representations. (2) Through the application of dependency parsing, a connection is established between entities and predicates. Following this, we incorporate the semantics of entities into the predicates, resulting in distinct weights being assigned to the relationships between various entities. We divide the question into two parts, the entity and the predicate, and then use different neural network methods to deal with these two parts, so our method is called the **C**hunked **L**earning **N**etwork (CLN).

This paper makes the following contributions:

- To address the distinctions in vector representation between entities and predicates, we employ separate modules for learning entities and predicates when tackling a question;
- By utilizing dependency parsing, we establish connections between entities and predicates, incorporating entity semantics into predicates to derive distinct weights for their relationships;

- The effectiveness of the CLN is demonstrated through experiments conducted on datasets containing both simple and complex questions.

This paper focuses on addressing the challenges in knowledge graph question answering (KGQA) by proposing an innovative solution. The introduction provides an overview of the current problems in KGQA and discusses existing methods, highlighting their limitations and unresolved issues. The related technologies section offers a comprehensive review of knowledge graphs, question-answering systems, and relevant methodologies. The proposed method section presents a novel approach, emphasizing head entity learning, relation learning, and their integration. The experimental evaluation section presents the experimental setup, dataset description, and performance analysis, demonstrating the superiority of the proposed method. This structured paper contributes to the advancement of KGQA by addressing challenges, introducing relevant technologies, proposing an innovative method, and validating its effectiveness through experiments.

## 2. Related Work

### 2.1. Question Answering over Knowledge Graphs

The Austrian linguist Edgar W. Schneider is credited with coining the term "knowledge graph" as early as 1972. In 2012, Google introduced their knowledge graph, which incorporates DBpedia, Freebase, and other sources. KGQA utilizes triples stored in the knowledge graph to answer natural language questions. Knowledge graphs usually represent knowledge in the form of triples. The general format of triples is (head entity, relation, tail entity), such as (Olympic Winter Games, Host city and the number of sessions, Beijing 24th), where "Olympic Winter Games" is the head entity, "Beijing 24th" is the tail entity, and "Host city and the number of sessions" is the relationship between the two entities. We use the lowercase letters h, r, and t to represent the head entity, relation, and tail entity, respectively, and (h, r, t) represents a triple in the knowledge graph. In previous work [16], transforming a multi-constraint question into a multi-constraint query graph was proposed. Since these multi-constraint rules require manual design and the rules are not scalable, this method does not perform well with large-scale knowledge graphs. Bordes A et al. [17] proposed a system that learns to answer questions using fewer multi-constraint rules to improve scalability. It uses a low-dimensional space to project the subgraph generated by the head entity for question answering. Then, it calculates the relevance score and determines the final answer by sorting. Likewise, so as not to be constrained by manual design rules, Bordes A et al. [18] developed a model that maps the questions to vector feature representations. A similarity function is learned during training to score questions and corresponding triples. The question is scored using all candidate triples at test time, and the highest-scoring entity is selected as the answer. But the vector representation of the question adopts a method similar to the bag-of-words model, which ignores the language order of the question (for example, the expressions of the two questions "who is George W. Bush's father?" and "Whose father is George W. Bush?" obtained by this method are the same, but the meanings of the two questions are obviously different). To focus on the order of words in the question, Dai Z et al. [19] use a Bidirectional Gate Recurrent Unit [20] (hereinafter referred to as Bi-GRU) to model the feature representation vector of the sentence and convert a simple single-fact QA question analysis into probabilistic questions. However, when the knowledge graph is incomplete, it is difficult to find the appropriate answer through probability. Based on the latest graph representation technology, Sun H et al. [21] described a method that extracts answers from subgraphs related to questions and linked texts, and they obtained good results. When the knowledge graph is incomplete, this method is effective, but external knowledge is not always obtainable. Recently, some works [22,23] used knowledge graph embedding to deal with question answering. With knowledge graph embedding, the potential semantic information can be retained, and the incompleteness of the knowledge graph can be handled. But the above methods model the problem and candidate relations separately without considering the word-level interactions between them, which may lead to local optimal results. Xie et al. [24] used

a convolution-based topic entity extraction model to eliminate the noise problem in the process of extracting entities. Qiu et al. [25] proposed a global–local attention relationship detection model, using a local module to learn the features of word-level interactions and a global module to capture the nonlinear relationship between the question and the candidate relationship in the knowledge graph. Zhou et al. [26] proposed a deep fusion model based on knowledge graph embedding, which combines topic detection and predicate matching in a unified framework, where the model shares multiple parameters for joint training at the same time.

### 2.2. Knowledge Graph Embedding

TransE [26] is the most representative method of knowledge graph embedding that represents entities and relations as low-dimensional vectors and learns to translate entities through relations, aiming to capture semantic relationships between them. TransH [27] tries to solve the limitations of TransE in handling complex relationships, such as one-to-many, many-to-one, and many-to-many, by allowing an entity to have different representations under different relationships. However, TransH still assumes that the entities and relationships are in the same semantic space, which restricts the representation capabilities of TransH to some extent. TransR [28] treats an entity as a combination of multiple attributes, and different relationships focus on different attributes of the entity. TransR assumes that different relationships have different semantic spaces. For each triplet, the entity should be projected into the corresponding relationship space first, and then the translation from the head entity to the tail entity should be established. Previous research, like [23], has shown that TransE performs better on Freebase [1].

Another algorithm commonly used for embedding knowledge graphs is ComplEx [29]. ComplEx is an extension of the TransE algorithm that models relationships as complex-valued embeddings. It represents entities and relationships as complex-valued vectors, where each component consists of a real part and an imaginary part. By using complex-valued embeddings, ComplEx is able to capture rich semantic interactions and multi-relational patterns in the knowledge graph. The scoring function of ComplEx is based on the Hermitian dot product, which measures the plausibility of a triple (head entity, relationship, tail entity). ComplEx has been shown to be effective in capturing both one-to-one and many-to-many relationships. DistMult [30] is a simplified variant of ComplEx that models relationships as diagonal matrices. In DistMult, each relationship is represented by a diagonal matrix, where the diagonal elements capture the interaction between the head and tail entities. DistMult assumes that relationships are symmetric and does not consider complex interactions. The scoring function of DistMult is based on the dot product between the head entity and relationship embeddings, followed by element-wise multiplication and summing of the resulting vector. DistMult is computationally efficient and performs well on knowledge graph completion tasks. RotatE is an algorithm designed specifically for knowledge graphs with symmetric relationships, such as family relationships or sibling relationships. It represents relationships as rotations in the complex plane. RotatE [31] assumes that the head and tail entity embeddings can be rotated by a specific angle to represent the relationship between them. The scoring function of RotatE calculates the element-wise circular correlation between the embeddings, and the plausibility of a triple is determined based on the resulting score. By explicitly modeling rotational patterns, RotatE is effective in capturing symmetric relationships in the knowledge graph.

### 2.3. Syntactic Analysis

How can the machine be made to accurately understand the semantics of natural language? Manning CD et al. [32] propose a convenient dependency analysis method. The method can give the basic forms of words, mark the structures of sentences according to phrases and grammatical dependencies, and discover relationships between entities. Sun K et al. [33] combined the above method with Graph Convolutional Networks [34] (hereinafter referred to as GCNs) and used them in aspect-level sentiment analysis. Ver-

berne et al. [35] added a reordering step to existing paragraph retrieval methods. When reordering, a ranking algorithm is used to calculate the question's score, and syntactic features are added to the question as weights. Arif et al. [36] used tree kernels (i.e., partial tree kernels (PTKs), subtree kernels (STKs), and subset tree kernels (SSTKs)) to consider the syntactic structure between them to solve the answer-reordering problem. Alberto et al. [37] calculated the similarity between trees based on the number of substructures shared between two syntactic trees and used this similarity to identify problems related to a new problem. To enhance downstream dependency analysis, a novel skeleton grammar has been proposed [38], which effectively represents the high-level structure of intricate problems. This lightweight formalization, along with a BERT-based parsing algorithm, contributes to the improvement of the analysis. For question-answering tasks, we make an improved dependency matrix better suited for concise and structured interrogatives by using it as the input of the GCN.

## 3. Chunked Learning Network

In this section, we first give an overall description of our model, introducing what the model contains and the role of each module. Section 3.2 introduces the architecture of the head-entity-learning model in detail, including the role of each component and the formulas involved. Section 3.3 introduces the architecture of the relation-learning model and the difference between this module and the head-entity-learning model. Section 3.4 introduces the structure of the pruning operation and the role of this module. Section 3.5 introduces the implementation process of the answer selection module.

### 3.1. CLN Overview

The CLN, as a deployable component, consists of four parts and trains the model through knowledge graph embedding. The main idea is shown in Figure 1. Consider the input question, "Which Olympic Winter Games was held in Beijing?" The pruning module identifies the entity "Olympics Winter Games" in the question, and the head-entity- and relation-learning modules learn their vector representations separately. Then, the learned vector representations are combined with the knowledge graph, and the answer selection module selects the closest fact triplet to return as the answer. When analyzing a question, our assumption is that it comprises a solitary entity. During data processing, for the convenience of subsequent processing, we designate consecutive entities as a single entity. The pruning operation marks the entity in the question to reduce the search space. Then, it learns the vector representation of the entity and relation in the question in the embedding space. Ultimately, a meticulously crafted joint distance metric is employed to locate the tail entity based on the $h + r \approx t$ equation, subsequently returning it as the answer.

### 3.2. Head-Entity-Learning Module

Given a question with length **L**, our goal is to restore the entity representation in the same space as the knowledge graph. The vector should represent the head entity of the question as closely as possible. We first map its **i** tokens to the sequence of the word-embedding vector $X = [x_1, \cdots, x_L]$. To take into account the sequential importance of words in the question, we use the Bidirectional Simple Recurrent Unit [39] (hereinafter referred to as Bi-SRU) to retain the global information of the question. Taking the forward direction as an example, Equations (1)–(4) show the details of calculating $\overrightarrow{h_i} \in \mathbf{R}^{d_h}$ where $d_h$ is the dimension of the hidden-state output of the Bi-SRU.

$$f_i = \sigma\left(W_{x_f x_i} + W_{h_f} \overrightarrow{h}_{i-1} + b_f\right) \tag{1}$$

$$r_i = \sigma\left(W_{x_r x_i} + W_{h_r} \overrightarrow{h}_{i-1} + b_r\right) \tag{2}$$

$$s_i = f_i \odot s_{i-1} + (1 - f_i) \odot W_{h_s} \overrightarrow{h}_{i-1} \tag{3}$$

$$\overrightarrow{h_i} = r_i \odot g(s_i) + (1 - r_i) \odot x_i \tag{4}$$

where $f_i$, $r_i$, and $s_i$ are the forget gate, reset gate, and internal state, respectively. $\sigma$ and $\mathbf{g}()$ represent activation functions, and $\odot$ denotes the Hadamard product.

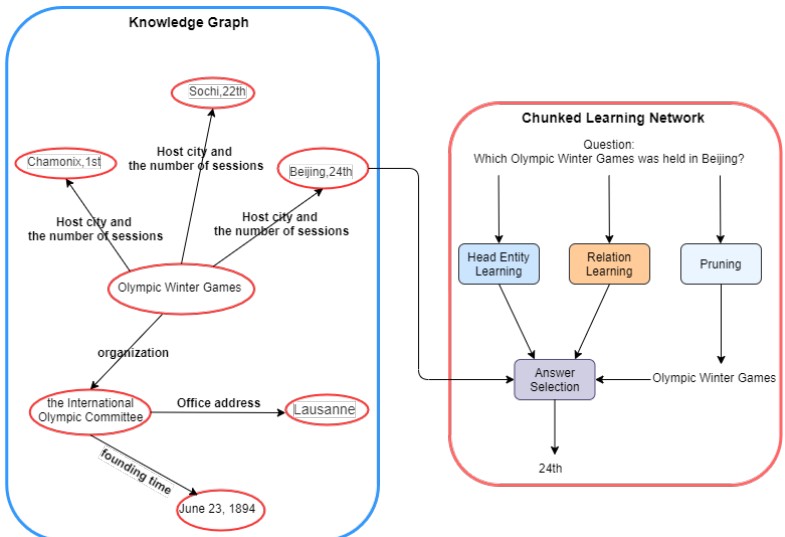

**Figure 1.** Overview of CLN. The **right** half shows the components of the model, and the **left** half is a simple schematic diagram of the knowledge graph.

By combining the hidden states from both the forward and backward directions, we obtain the concatenated representation, denoted by $h_i = \left[ \overrightarrow{h_i}; \overleftarrow{h_i} \right]$. The Bi-SRU is complemented by a convolutional neural network (hereinafter referred to as CNN) module, which captures nearby contextual information that is in proximity to the entity. Equation (5) shows the calculation process of the $j$-th feature map of the $l$-th layer.

$$c_j^l = ReLU \left( \sum_{i=1}^{L} c_i^{l-1} * k_{ij}^l + b_j^l \right) \tag{5}$$

where $c_i^{l-1}$ represents the $i$-th input of the $(l-1)$-th layer (when $l = 1$, $c_i^{l-1} = h_i$), the symbol $*$ represents the convolutional operation, $k_{ij}^l$ represents the weight of convolution kernel $j$ corresponding to the $i$-th input feature, and $b_j^l$ is the bias of the convolution kernel. In the network described in this paper, we employ the rectified linear unit (ReLU) to compute feature maps.

After the convolution operation, we replace the pooling layer with an attention layer and apply its result to $h_i$. Equations (6) and (7) illustrate this process. The weight and bias of this layer are denoted by $w$ and $b$, respectively. In this way, not only can the information of the entity be extracted, but the contextual information can also be preserved to a certain extent.

$$\alpha_i = \frac{exp(c_i)}{\sum_{i=1}^{L} exp(c_i)} \tag{6}$$

$$e_i = tanh\left(w_e^T \alpha_i h_i + b_e\right) \tag{7}$$

The result $e_i$ is then used as the target vector of the *i*-th token, and Equation (8) represents using the average of the target vectors of all tokens as the predictive representation of the entity.

$$\widehat{e_h} = \frac{1}{L} \sum_{i=1}^{L} e_i^T \tag{8}$$

$\widehat{e_h}$ represents the learned entity vector representation. We independently train this module so that the vector representations of entities in sentences are as close as possible to the representations of entities in triples. The head-entity-learning module of the CLN is depicted in Figure 2.

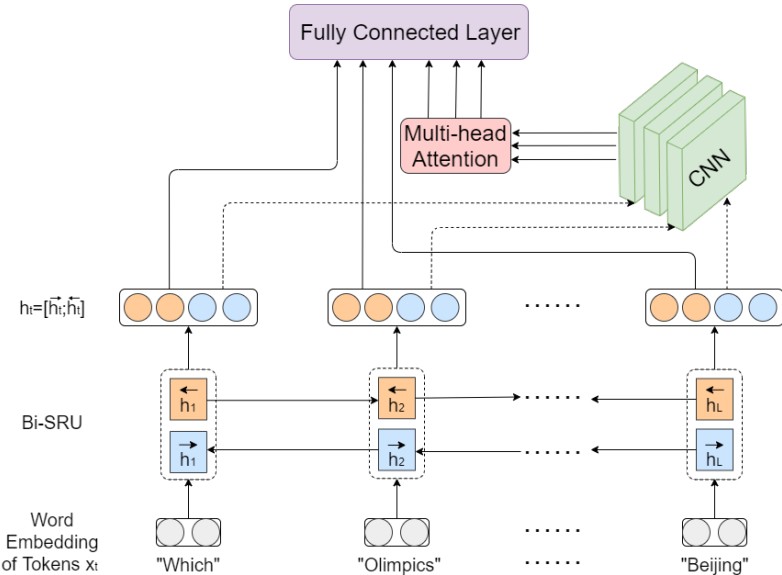

**Figure 2.** Proposed head-entity-learning module.

### 3.3. Relation-Learning Module

For relation-modeling problems, semantic parsing is the foundation of traditional methods of mapping relationships or for dictionary construction. Since the user's questions are not restricted, the predicates for the new questions may differ from the predicates in the training data. We therefore use the global information preserved in the knowledge graph embedding space to represent relational information in the question.

Similar to Equations (1)–(4), we obtain the vector representation $h_i \in \mathbf{R}^{d_{2h}}$ of the sentence through the Bi-SRU. Because the context of entities often contains relation features, and entities typically have similar features and attributes, the CNN extracts features from local regions by using convolutional kernels with shared weights. Conversely, as relationships between entities tend to be non-local and encompass the entire graph, the GCN can aggregate information from neighboring nodes and propagate this information to other nodes in the graph, thereby modeling complex relational patterns. To maximize the effect of dependency analysis, we construct the result into a dependency matrix as the adjacency matrix of the graph to spread node feature information.

The same predicate is associated with different entities, each making distinct contributions, so we capture the dependency between the predicate and the entity through syntactic analysis to obtain the adjacency matrix $\mathbf{A} \in \mathbf{R}^{L \times L}$. The adjacency matrix of each word and itself is set to self-loop; that is, the diagonal value of $\mathbf{A}$ is 1. These constraints between words are aggregated through a graph convolutional layer into a vector representation of predicates. With these constraints, different vector representations can be produced by the

same predicate when connecting different entities. The calculation process of feature fusion is shown by Equations (9) and (10).

$$\widetilde{g}_i^l = \sum_{j=1}^{L} A_{ij} W^l g_j^{l-1} \tag{9}$$

$$\widetilde{g}_i^l = ReLU\left(\frac{\widetilde{g}_i^l}{d_i+1} + b^l\right) \tag{10}$$

where $g_j^{l-1} \in R^{2d_h}$ is the representation of the $j$-th token obtained from the previous GCN layer (when $l = 1, g_j^{l-1} = x_j$), $g_i^l \in R^{2d_h}$ is the output of the current GCN layer, $d_i = \sum_{j=1}^{L} A_{ij}$ is the degree of the $i$-th token in the dependency tree, and $W^l$ and $b^l$ are the weight and bias matrices in the GCN layers, respectively.

Similar to Equations (6)–(8), we fuse the output of the GCN layer to $h_i$ through the attention mechanism as the representation of a single token, and the mean of all representations is the vector representation of the relation. Equations (11)–(13) show the details of the calculation process.

$$\beta_i = \frac{exp\left(\sum_{i=1}^{L} g_i^l\right)}{\sum_{i=1}^{L} exp\left(\sum_{i=1}^{L} g_i^l\right)} \tag{11}$$

$$p_i = tanh\left(w_p^T \beta_i h_i + b_p\right) \tag{12}$$

$$\widehat{p}_l = \frac{1}{L}\sum_{i=1}^{L} p_i^T \tag{13}$$

Through independent training, we ensure that the relation representation is as close as possible to the representation of the relation in the triplet. Figure 3 shows the relation-learning module of the CLN. As shown in Equation (14), both head-entity-learning and relation-learning modules use MSEloss as the loss function during head entity training, where $\widehat{p}_l = \widehat{e}_h$, and $p_l = e_h$.

$$loss(\widehat{p}_l, p_l) = (\widehat{p}_l - p_l)^2 \tag{14}$$

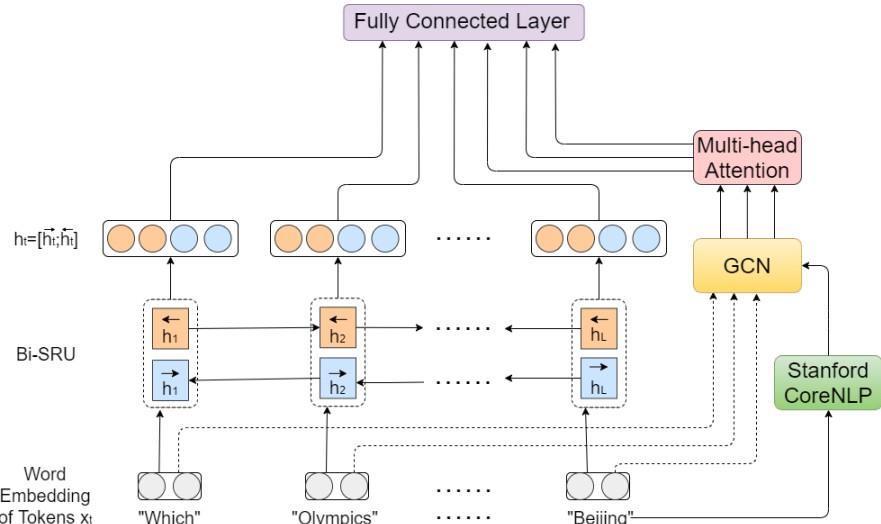

**Figure 3.** Proposed relation-learning module.

### 3.4. Pruning Module

In Section 3.1, we introduced the objective of this module, which is to recognize entities by identifying one or more consecutive words in the question. This enables us to designate the entire search space as a collection of multiple entities sharing similar or identical names. To streamline the module, we employ "Bi-LSTM (Bidirectional Long Short-Term Memory) + softmax" exclusively for named entity recognition. The question **q**, along with its associated entities, serves as the training data for the pruning model. Given that the topic entity in the question is contiguous, the model will identify the continuous words in the test set as either entities or as components of the correct topic entity. Hence, all words that are identical to the topic entity or contain the topic entity will be treated as candidate entities. The entity and non-entity tokens ($HED_{entity}$ and $HED_{non}$) obtained will be passed to the answer selection module.

### 3.5. Answer Selection Module

This module receives the output of the first three modules as input and finds the answer from the knowledge graph based on the principle of $f(e_h, p_l) \approx e_t$. The above process is achieved by using the joint distance metric proposed in [23]. A candidate fact is a fact triple whose head entity belongs to the candidate head entity. Consider **C** as the set comprising all candidate facts.

$$minimize_{h,l,t \in C} \|p_l - \hat{p_l}\|_2 + \beta_1 \|e_h - \hat{e_h}\|_2 + \beta_2 \|f(e_h, p_l) - \hat{e_t}\|_2 - \beta_3 sim[n(h), HED_{entity}] - \beta_4 sim[n(l), HED_{non}] \quad (15)$$

As shown in Equation (15), $p_l$ and $e_h$ are the relation and entity embeddings in the knowledge graph, respectively. The $sim[x, y]$ function measures the similarity between two strings, and $n(x)$ returns the name of an entity or predicate. $\beta_1$, $\beta_2$, $\beta_3$, and $\beta_4$ are predefined weights used to balance the contribution of each item.

## 4. Experiments and Analysis

In this section, we first verify the effectiveness of our model on public datasets and then analyze the reasons why our model can improve accuracy.

### 4.1. Datasets

The knowledge graphs and datasets used in the experiments can be downloaded through public channels.

FB5M [1]: The data in Freebase contain a lot of topics and types of knowledge, including information about humans, media, geographical locations, and so on. In our study, we utilized FB5M, which is among the more expansive subsets of Freebase.

SimpleQuestions [7]: This dataset comprises over 10,000 Freebase-related questions, with the issues within the dataset being summarized using facts and articles as references.

FB5M was employed as the knowledge graph in our study, and TransE was used for knowledge graph embedding to learn entity and relation representations. The performance of the model is measured by the accuracy of finding the ground truth.

### 4.2. Overall Results

Now, we will discuss the performance of the CLN. We take several representative KGQA methods as baseline models and compare the results. These works include the Bi-GRU rank model from Dai et al. [19], the Memory Network approach from Bordes et al. [7], the character-level CNN from Yin et al. [40], the character-level encoder–decoder from Golub et al. [41], the KG embedding method from Huang et al. [23], and the transformer-based question encoder from Li et al. [42]. Table 1 presents quantitative disparities in the performance of various methods on the dataset.

**Table 1.** Performance of different methods on SimpleQuestions.

| Methods | Accuracy |
|---|---|
| Cfo [19] | 0.626 |
| MemNNs [7] | 0.639 |
| AMPCNN [40] | 0.672 |
| Character-level [41] | 0.703 |
| KEQA [23] | 0.749 |
| Te-biltm [42] | 0.751 |
| **CLN (ours)** | **0.753 (+11.4%)** |

Note: Since the Freebase API is no longer available, thanks to Huang et al. [23] for re-evaluating the Freebase-API-based models of Cfo [19] and AMPCNN [40] for new results.

Accuracy is calculated by comparing the predicted entity–relationship pair with the ground truth, and the result is considered correct only when the entity–relationship pair given by the model conforms to the ground truth. From the results in Table 1, we can see that our model outperforms the previous methods. Compared with the accuracy when SimpleQuestions was released [7], the accuracy of the CLN was improved by 11.4%. This is due to a more complex neural network design and the use of different models for different subtasks.

*4.3. Comparison of Baselines*

To represent the baseline models more clearly, according to the results in Table 1, we introduce the differences between the CLN and the baseline models and then explain the reasons for the improved accuracy.

Cfo [19]: A Bi-GRU is used to rank candidate predicates. When the knowledge graph is incomplete, it is difficult to obtain the ranking through the established probability model. Our model uses knowledge map embedding, which can have a similar vector representation of words with similar meaning.

MemNNs [7]: This approach acquires entity and predicate representations from training questions and compares the entities and predicates within the new question with the previously acquired vectors. However, this method requires a large amount of data to train the classifier, and we use different models to train entities and predicates, which will improve the accuracy with the same amount of data.

AMPCNN [40]: This method uses a character-level CNN to match topic entities in fact candidates with entity descriptions in questions. However, the description of the relationship may not be limited to a part of the text, long-distance dependencies may exist, and the length is inconvenient to estimate. So, we use the GCN to capture long-distance dependencies for better accuracy.

Character-level [41]: The authors designed a character-level encoder–decoder frame, where each character corresponds to a one-hot vector. This makes the parameters of the model larger and uses more resources. We use word-level encoding to guaranteed accuracy without using pre-trained language models.

KEQA [23]: Using KG embedding, the model learns how to express the entity and predicate of the question. On this basis, we use different models to deal with entities and relations, making the model targeted.

Te-biltm [42]: The authors changed the encoder part of the transformer to Bi-LSTM, which can make the encoding of words obtain directional information for relation extraction in question answering. We are not limited to a single subtask but use different models to complete different subtasks, and the results from each module are fused to obtain the answer to the question, which is also the theme of this paper.

4.3.1. Statistical *t*-Test

We performed a statistical significance test using SPSS software to validate the results of our proposed method. Our objective was to assess the statistical significance of the

accuracy achieved through our approach. The t-test was employed to generate a p-value, which measures the probability of the observed results being due to chance. A lower *p*-value indicates a higher likelihood of statistical significance. To evaluate our method, we compared the results obtained from the CLN-based model with those presented in Table 1. Notably, the accuracy achieved yielded a *p*-value of 0.036. These findings substantiate the statistical significance of the improvements attained by our proposed method.

### 4.3.2. Ablation Study

We removed different parts of our model and verified statistical significance, and the results obtained are shown in Table 2. Due to the effect of the CNN and multi-head attention, the accuracy of our head-entity-learning model is improved by 0.4%. In addition, the use of the Bi-SRU shortens the training time. As mentioned in the previous paragraph, the use of more complex models increases the accuracy of relation learning by 0.4%. The improvements of these two models have improved the final accuracy. We can see that there is a statistically significant improvement over the baseline when both CLN modules exist at the same time.

**Table 2.** Ablation experiment results.

| Head-Entity-Learning Module, Accuracy | Relation-Learning Module, Accuracy | Total Accuracy |
|---|---|---|
| Bi-GRU + attention, 0.644 | Bi-GRU + attention, 0.815 | 0.749 |
| CLN_entity, 0.647 | CLN_relation, 0.818 | 0.753 |
| Bi-GRU + attention, 0.644 | CLN_relation, 0.818 | 0.751 |
| CLN_entity, 0.647 | Bi-GRU + attention, 0.815 | 0.752 |

### 4.3.3. Qualitative Analysis

Different models obtain the embeddings of the same sentence at the same epoch, as shown in Figure 4. The scatter plot indicates that the enhanced model effectively discerns the distinctions between words, and points with similar values represent the predicate fusion with the representation of the entity. It is not difficult to find that after the "Bi-GRU + attention" sentence representation, the model needs a lot of data to learn weights. The improved model only needs a small number of samples to learn the vector representation of sentences better and faster.

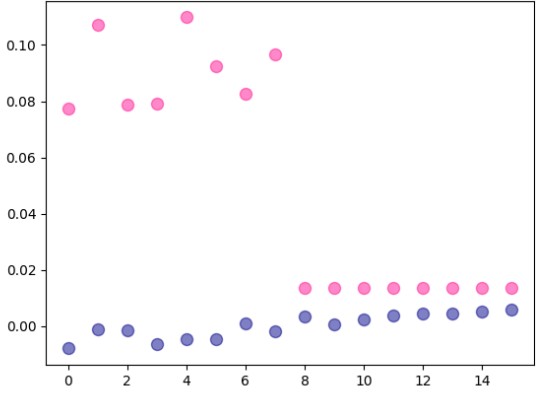

**Figure 4.** Result analysis of different modules. The horizontal axis refers to the words in the sentence, the vertical axis refers to the vector representation of the word, and the right half of the sentence represents the words that are padded to make the sentences the same length. The relationship-learning module's sentence representation with the CLN is represented by pink dots, while the "Bi GRU+attention" sentence representation is depicted by blue dots.

We delve into the joint impact of semantic parsing and the GCN using accuracy as an example. In Figure 5, we can see that in the initial stage of training, the accuracy of the relation-learning module rises rapidly, thanks to the combined effect of semantic parsing and the GCN. When the relationship between words in all sentences is constructed, the change in accuracy is relatively smooth. This trend of the change can also be seen through the change in the loss of the head-entity-learning module. But this method is only suitable for relational construction, so we only use this feature in relation-learning models.

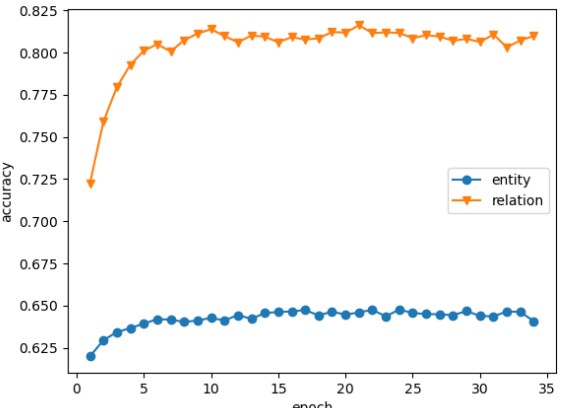

**Figure 5.** Variation in accuracy of head-entity-learning module and relation-learning module.

From the loss curves of the two modules in Figure 6, we can see that the loss of the head-entity-learning module decreases rapidly at the beginning of training and then tends to be flat, which indicates that the module has achieved good performance. At the same time, the relation-learning module loss drops rapidly and remains largely unchanged in the following periods, indicating that when the relational construction of words in all sentences is completed, other parts of the model can also support relational learning well.

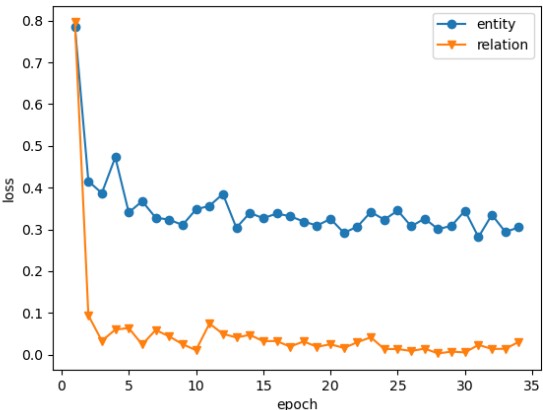

**Figure 6.** Variation in loss of head-entity-learning module and relation-learning module. In order to facilitate observation, we expanded the value of loss by 1000 times.

### 4.3.4. Error Analysis

In cases where the knowledge graph does not contain the required information to answer the question, the model cannot implement question answering. Most of the time, it is because the answer to the question is not unique. For example, consider the question, "Which actor was born in Warsaw?" Experiments show that the model can correctly learn the vector representation of "Warsaw" in the head-entity-learning module and "location.location.people_born_here" in the relation-learning module. But only one item,

$\|f(e_h, p_l) - \hat{e}_t\|_2$, in the answer selection module involves tail entities, so it cannot represent a large set of answers.

### 4.3.5. Component Introduction Experiment

The purpose of this experiment is to assess the viability of our proposed model, which has a small parameter count and can be seamlessly integrated as a component into other models. Our objective is to verify the applicability of our method on different datasets. We conducted experiments using the WebQuestion dataset and introduced the proposed component into two existing models: EmbedKGQA [22] and TransferNet [43]. In EmbedKGQA, we incorporated the results of the relation-learning module into the inference module. For TransferNet, we introduced the outputs of the head-entity-learning module and the relation-learning module into step t using an attention mechanism. The results obtained are shown in Table 3.

**Table 3.** Hits@1 results on WebQuestionsSP.

| Methods | WebQuestionsSP |
|:---:|:---:|
| EmbedKGQA | 66.6 |
| EmbedKGQA + CLN | 67 |
| TransferNet | 71.4 |
| TransferNet + CLN | 71.6 |

The diverse architectures and parameter settings of different models can lead to variations in the performance of the introduced component within each model. A component that exhibits promising performance in one model may not achieve its optimal effectiveness when placed in another model. Furthermore, the design and functionality of other components within the model can also impact the performance of the introduced component. If there is a close interaction or dependency between the other components in the model and the specific component being introduced, placing that component in different models may yield different effects on its performance. It is worth noting that both our proposed model and EmbedKGQA leverage knowledge graph embeddings. Thanks to the shared utilization of knowledge graph embeddings, which enhances the models' ability to capture semantic relationships and facilitate reasoning capabilities, the introduced component exhibits enhanced effectiveness when integrated into our model and EmbedKGQA.

### 5. Conclusions

We propose a Chunked Learning Network for KGQA in this paper. The objective is to address the challenge of machines struggling to comprehend the semantic meaning of a question. The model incorporates the vector representation of entities and predicates into the question by utilizing the knowledge graph embedding. It employs distinct processing methods for different word types within the question. Words with similar meanings, such as word abbreviations, exhibit similar vector representations within the vector space. Additionally, the graph convolutional neural network assigns varying weights to capture the dependency relationship between words, thereby enhancing the contextual impact on each word. The experimental results demonstrate that our method enhances KGQA accuracy on datasets, and the proposed components indicate a promising direction for future research. However, our method currently falls short in entity recognition accuracy and faces challenges in coping with the expanding knowledge graph. To overcome this challenge, we plan to take into account the dynamic properties of the knowledge graph, as they are frequently updated in real-world scenarios.

**Author Contributions:** Writing—original draft, Z.Z. (Zicheng Zuo); Writing—review & editing, Z.Z. (Zhenfang Zhu), W.W. (Wenqing Wu), W.W. (Wenling Wang), J.Q. and L.Z. All authors have read and agreed to the published version of the manuscript.

**Funding:** This research received no external funding.

**Data Availability Statement:** The data that support the findings of this study is openly available at https://github.com/ZuoZicheng/CLN, accessed on 18 July 2023.

**Conflicts of Interest:** The authors declare no conflict of interest.

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
