# Peer review of "Improving Question Answering over Knowledge Graphs with a Chunked Learning Network"

_electronics, doi:10.3390/electronics12153363_

Round 1

Reviewer 1 Report

Natural Language Processing (NLP) is acquiring a large amount of attention, given the fact that nowadays many query architecture are based in NLP for a more intuitive, flexible, dynamic, and friendly interaction with the human user.

Nevertheless, there is a number of issues regarding the efficient and effective processing of the user's query as well as the returned answer. Artificial Neural Networks (ANN) based architectures offer promising results in this field, and they are being applied in many real world problems.

Finally, an additional issue is related to the way in which the knowledge is stored, in order to be properly accessed by the user.

Authors propose a model called Chunked Learning Network (CLN), in the sense that it manages the knowledge represented by graphs in chunks, allowing a better performance in the query process. The paper is well structured and offers enough introductory material for the reader, as well as a definition of the main components of the model. Experimental results are discussed in depth, by the comparison of the proposed method with existing ones.

The results are interesting and promising, and the methodology is well described, as long as I understand. There are some minor issues regarding the writing and presentation. For example, bibliographical references should follow a common format and style. In page 3, papers from the same authors are cited in different way ("A Bordes" and "Bordes A"). The references section shows that not all the entries follow the same style.

There are some minor issues regarding the writing in the document:

Page 4, line 177: "hereinafer" instead of "hereinafter".

Page 10, line 339: "They uses" instead of "They use".

Page 10, line 334: "They designs" instead of "They design".

Reviewer 2 Report

Thank you for the opportunity to review this article. The authors of this paper investigate KGQA, which is an interesting topic. However, there are several issues that the authors need to address to further improve the paper, which I detail below.

1. The authors need to improve the introduction.

The introduction needs to be constructed based on research problem. I missed a robust construction KGQA and the development of research questions. The paper does not make clear the problem to be investigated. Is it KBQA or KGQA? Give some more info on the relationship between them. The authors state that “the paper focuses on addressing the challenges in KBQA by proposing an innovative solution”. Why KBQA instead of KGQA which is your research goal? Give some examples to elucidate this point. What is the importance of this study? To whom? Try to bring what has not been studied, based on previous researches. What is the contribution of previous researches on this topic? Which methods do they use to approach the topic (KGQA)? Try to find the gap and the your contribution.

A conceptual framework in grahical representation would be helpful for readers (what is missing, how are you going to solve it, which methods do you employ, what is the outcome).  

2. The authors are encouraged to strengthen the conclusion part. 

What is your contribution to the KGQA? Is your research helpful to the end users and the academic community and how?  Despite your valued efforts and results, are there any limitations? In general, you need to provide a theoretical contribution. The theoretical contribution focuses on explain your findings and should address how your findings further extend existing knowledge in the field, by providing more details.

3. There paper needs to be edited thoroughly. For instance, in lines 20-25 or in figure 1 and 4. Specifically, in figure 1, you provide some important info that should be moved into the main part and not in the title of the figure. 

Reviewer 3 Report

The paper contributes to the improvement of knowledge graph question answering proposing  a novel method - Chunked Learning Network.

Strengths:

• an innovative method is introduced for question answering;

• separate modules are used for learning entities and predicates when tackling a question, in order to address the distinctions in vector representation between entities and predicates

dependency parsing is used to establish connections between entities and predicates;

• the effectiveness of the proposed method is demonstrated through experiments by using datasets that contain both simple and complex questions;

• the paper is well written and clear; 6 figures and 3 tables illustrate the discussion;

future work is convincing.

Minor editing is needed.

Author Response

We would like to extend our heartfelt gratitude for your thoughtful and meticulous review of our manuscript. Your insightful comments and suggestions have been immensely valuable in shaping the final version of our paper.